# Manganese exposure in juvenile C57BL/6 mice increases glial inflammatory responses in the substantia nigra following infection with H1N1 influenza virus

Collin M. Bantle[1], C. Tenley French[1], Jason E. Cummings[2], Shankar Sadasivan[3], Kevin Tran[1], Richard A. Slayden[2], Richard J. Smeyne[3], Ronald B. Tjalkens[1]*

1 Department of Environmental and Radiological Health Sciences, Colorado State University, Fort Collins, Colorado, United States of America, 2 Department of Microbiology, Immunology and Pathology, Colorado State University, Fort Collins, Colorado, United States of America, 3 Department of Neuroscience, Vickie & Jack Farber Institute for Neuroscience, Thomas Jefferson University, Philadelphia, Pennsylvania, United States of America

* ron.tjalkens@colostate.edu

**Data Availability Statement:** All relevant data are within the paper and its Supporting information files. The Sequence Read Archive (SRA) NCBI

## Abstract

Infection with Influenza A virus can lead to the development of encephalitis and subsequent neurological deficits ranging from headaches to neurodegeneration. Post-encephalitic parkinsonism has been reported in surviving patients of H1N1 infections, but not all cases of encephalitic H1N1 infection present with these neurological symptoms, suggesting that interactions with an environmental neurotoxin could promote more severe neurological damage. The heavy metal, manganese (Mn), is a potential interacting factor with H1N1 because excessive exposure early in life can induce long-lasting effects on neurological function through inflammatory activation of glial cells. In the current study, we used a two-hit model of neurotoxin-pathogen exposure to examine whether exposure to Mn during juvenile development would induce a more severe neuropathological response following infection with H1N1 in adulthood. To test this hypothesis, C57BL/6 mice were exposed to $MnCl_2$ in drinking water (50 mg/kg/day) for 30 days from days 21–51 postnatal, then infected intranasally with H1N1 three weeks later. Analyses of dopaminergic neurons, microglia and astrocytes in basal ganglia indicated that although there was no significant loss of dopaminergic neurons within the substantia nigra pars compacta, there was more pronounced activation of microglia and astrocytes in animals sequentially exposed to Mn and H1N1, as well as altered patterns of histone acetylation. Whole transcriptome Next Generation Sequencing (RNASeq) analysis was performed on the substantia nigra and revealed unique patterns of gene expression in the dual-exposed group, including genes involved in antioxidant activation, mitophagy and neurodegeneration. Taken together, these results suggest that exposure to elevated levels of Mn during juvenile development could sensitize glial cells to more severe neuro-immune responses to influenza infection later in life through persistent epigenetic changes.

accession number for the RNA-Seq files is:
Accession number PRJNA684630 NCBI Link:
https://www.ncbi.nlm.nih.gov/Traces/study/?acc=
PRJNA684630&o=acc_s%3Aa.

**Funding:** RBT - ES030937 National Institutes of
Health nih.gov No role in study design, data
collection or analysis, decision to publish, or
preparation of the manuscript.

**Competing interests:** The authors have declared
that no competing interests exist.

## Introduction

Parkinson's disease (PD) is characterized by the loss of voluntary motor control due to the
degeneration of dopaminergic neurons in the substantia nigra pars compacta (SNpc) with
associated α-synuclein protein-aggregation, neuroinflammatory activation of glial cells, mito-
chondrial dysfunction and oxidative stress [1]. Although genome-wide association studies
(GWAS) have identified genetic variants in familial forms of the disease, these represent a very
small percentage of individuals with PD, with the majority of PD cases thought to be sporadic
or of an unknown etiology [2]. Epidemiological and experimental evidence suggests that envi-
ronmental neurotoxin exposure and viral infections are possible risk factors for sporadic PD.
Following the 1918 "Spanish Flu" pandemic, nearly every patient who had an acute episode of
encephalitis lethargica (EL) from the H1N1 infection went on to develop postencephalic par-
kinsonism and individuals who were born between 1888 and 1924 had a two to three-fold
higher risk of developing Parkinson's disease later in life than those born outside of that range
[3–6].

   Our lab and others have recently shown that exposure to certain classes of enveloped RNA
viruses, Western equine encephalitis virus (WEEV) and H5N1 (strain, A/VN/1203/04) via
intranasal infection can induce loss of dopaminergic neurons in the SNpc [7, 8]. Infection with
viruses such as H5N1 avian influenza virus, WEEV and H1N1 induce neuronal loss in part
through the activation of microglia and astrocytes and subsequent release of glial-derived neu-
rotoxic inflammatory mediators [7–9]. Microglia and astrocytes express a variety of damage-
associated molecular patterns (DAMPs) that promote inflammation and disease progression
in postencephalic parkinsonism [10]. Reactive microglia and astrocytes have increasingly
become the focus of studies examining the pathophysiology of PD, suggesting that neuroin-
flammation may be a link between viral encephalitis and the development of parkinsonian
neurological symptoms [11]. Additionally, astrocytes and microglia have innate immunologi-
cal memory in the brain to facilitate a rapid inflammatory response to recurrent inflammatory
stressors, and it has been postulated that this acute and exacerbated inflammatory response
from glia may have the capacity to exacerbate neuronal injury following secondary insults
[12].

   How encephalitic infections from non-neurotropic H1N1 virus cause neurological dysfunc-
tion and neurodegeneration in certain individuals is not entirely clear but may be related to
the severity of the neuroinflammatory response [7, 8, 13–17]. This could be due to prior expo-
sure to environmental neurotoxins that activate innate immune inflammatory signaling in
microglia and/or astrocytes, thereby sensitizing the tissue environment of the brain to greater
inflammatory activation of glial cells during an encephalitic infection. Data from recent studies
examining innate immune memory in microglia support this possibility, where multiple injec-
tions of LPS in mice resulted in immune training that amplified activation of microglia and
astrocytes as well as levels of inflammatory cytokines upon later immunological challenge [12].
Similarly, mice infected with H1N1 prior to treatment with the neurotoxin, 1-methyl-4-phe-
nyl-1,2,3,6-tetrahydropyridine (MPTP) had significantly greater activation of microglia and
loss of dopamine neurons in the SNpc that mice treated only with MPTP [14]. Elevated levels
of Mn during juvenile development in mice can exacerbate neuroinflammatory activation of
glia and increase the production of reactive oxygen and nitrogen species and inflammatory
cytokines following a subsequent exposure to Mn [18]. Rats exposed to Mn during juvenile
development at concentrations comparable to low environmental exposure levels did not have
overt neurotoxicity but showed evidence of deficits in mitochondrial respiration, as well as oxi-
dative stress and chronic neuroinflammation [19]. Both Mn and viral infection induce inflam-
matory activation of microglia and astrocytes and mitochondrial stress in PD-relevant brain

regions, suggesting that neuroinflammatory responses to Mn and encephalitic infections may evoke common pathophysiological signaling mechanisms in glial cells leading to altered innate immune responses in the brain [18, 20, 21].

Previous work suggests that glial inflammatory responses to Mn can exacerbate neuronal injury following secondary neurotoxic insults [10, 18, 22–25] but whether Mn can also enhance the severity of neurological damage from encephalitic viral infections with H1N1 is unknown. In the current study we investigated whether elevated levels of an Mn during juvenile development could enhance neuroinflammatory damage to dopaminergic neurons after infection with H1N1 influenza virus and increase one's risk of developing neurological disease later in life. We tested this hypothesis by exposing juvenile C57BL/6 mice to $MnCl_2$ in drinking water (50 mg/kg/day) for 30 days from days 21–51 PN, followed by intranasal infection with H1N1 at PN72. Control mice received only drinking water followed by either mock infection or infection with H1N1. Stereological counts of dopaminergic neurons and microglia in the SNpc were performed. We noted pronounced microglia activation following dual treatment with Mn and H1N1 relative to either treatment alone, as well as marked astrogliosis and increased in the number of reactive A1 astrocytes. RNA sequencing (RNAseq) analysis revealed activation of multiple stress response pathways involved in antioxidant activity, mitophagy, anti-viral activity and neurodegeneration in mice treated with Mn and subsequently exposed to H1N1. Collectively, these findings suggest that exposure to elevated levels of Mn during juvenile development increases neuroinflammatory activation of glia following encephalitic infection with H1N1 influenza virus later in life, likely through epigenetic modification of histones that increases secondary innate immune responses in microglia and astrocytes.

## Materials and methods

### Exposure protocol with manganese and H1N1

All procedures were approved by Colorado State University and St. Jude Children's hospital Institutional Animal Care and Use Committee (IACUC) and were conducted in compliance of the National Institute of Health guidelines. Dosing was performed as previously published [26]. C57Bl/6 mice were obtained from the Jackson Laboratory and housed in a temperature-controlled room (maintained at 22–24°C on a 12 hr light/dark cycle) with *ad libitum* access to standard chow. At day P21, male and female C57BL/6 mice were administered $MnCl_2$ (50mg/kg/day; Sigma) or normal drinking water. The dose of $MnCl_2$ was calculated by monitoring water intake and weight gain for thirty days, with the concentration in drinking water adjusted to deliver 50 mg/Kg/day according to water consumption. At P51, Mn-treated mice were placed back on regular drinking water for a period of one month. Mice were then intranasally infected with A/California/04/2009 (CA/09) H1N1 or mock-infected with saline. Infection with H1N1 was performed as described previously [27]. Briefly, for infections, mice were lightly anesthetized with isofluorane and intranasally inoculated with either $10^3$ $TCID_{50}$ of CA/09 in 25μl of phosphate-buffered saline (PBS) or PBS alone and monitored daily for 21 days post-infection by assessing any possible neurobehavioral abnormalities or clinical signs of illness. A clinical scoring system was used to identify any animals with untoward morbidity, the treatments performed did not produce any overt morbidity and all animals maintained normal body weight and feeding behavior, relative to untreated/mock-infected controls. Clinical scoring was performed by laboratory staff and was supported by daily observation from dedicated laboratory animal veterinary personnel. After 21 days, infected or mock-infected control mice were euthanized for tissue collection. Euthanasia was performed under deep isofluorane anesthesia.

## Preparation of A/California/04/2009 (CA/09) H1N1 inoculum

A detailed description of inoculum preparation and administration has been previously reported by our group [27]. In brief, A/California/04/2009 (CA/09) H1N1 virus was passaged in the allantoic cavity of 10-day-old specific pathogen-free embryonated chicken eggs. At 48 to 72 hours post-infection, allantoic fluid was harvested, clarified by centrifugation, and stored at −70˚C. Tissue culture infectious dose 50% ($TCID_{50}$) titers were determined using Madin-Darby canine kidney (MDCK) cells and evaluated by the method of Reed and Muench [28].

## Tissue processing for immunohistochemistry and immunofluorescence

Mice were anesthetized with Avertin and transcardially perfused with 4% paraformaldehyde in PBS. Brains were dissected and processed for paraffin embedding. Brains were then sectioned on the microtome at 10µm thickness and mounted on polyionic slides (Superfrost-plus, Fisher Scientific). Deparaffinized SN sections were incubated with primary antibody for identification of dopaminergic neurons [mouse monoclonal anti-tyrosine hydroxylase (TH; Sigma-Aldrich; 1:500), microglia [rabbit polyclonal anti-IBA1 (Wako Chemicals; 1:500), astrocytes [rabbit polyclonal anti-S100beta (Abcam; 1:500)], complement C3 [rat monoclonal anti-C3 (Abcam; 1:100)], SerpinA3 [mouse monoclonal anti-SerpinA3 (thermos; 1:250)], IP-10 [mouse mono-clonal anti-IP-10 (Santa Cruz; 1:500)], IP-10/Cxcl10 [mouse monoclonal anti-cxcl10 (Santa Cruz; 1:250)], CCL2 [mouse monoclonal anti-Ccl2 (Millipore; 1:500)], and acetylated Lysine residues [rabbit polyclonal anti-acetylated lysine (Cell Signaling; 1:500)]. For immunohisto-chemical analysis, the secondary antibodies included biotinylated mouse IgG (for TH, 1:1000) or biotinylated rabbit IgG (for IBA1, 1:1000). Diaminobenzidine (DAB) or a VIP kit (Vector labs) reaction was used to yield a brown (TH) or a purple (IBA1) color, respectively. For immunofluorescence, anti-mouse, anti-rabbit or anti-rat IgG alexa flour 555, alexa flour 488, or alexa flour 647 were diluted in TBS (2% Triton) at 1:500. Sections were washed 5X (5 min) and stained with DAPI in the final wash after an hour incubation period in the secondary anti-body. Sections were then mounted with medium, coverslipped and stored at 4˚C until imaged.

## Quantification of TH+ dopaminergic neurons and Iba1+ microglia in the substantia nigra pars compacta

Quantitation of neurons and glial cells was performed as previously reported [27]. In brief, TH + dopaminergic neurons and IBA1+ microglia in the SNpc were estimated using standard model-based stereological methods [16, 29]. Counts of total dopaminergic neurons and activated microglia were estimated using Microbrightfield StereoInvestigator (MBF Biosciences, Williston, VT) and the optical fractionator method using an Olympus BX-51 microscope and 100X objective [30, 31]. The identification of resting and activated microglia was based on defined morpho-logical criteria, as previously reported [16, 30]. Resting microglia were defined as having a small, oval IBA1+ cell body that averaged 3 microns in diameter with long slender processes, while microglia were classified as activated when the cell body was slightly increased in size compared to resting microglia and had an irregular shape, with shorter and thickener processes. The investigator was blinded from all experimental groups during imaging and cell quantitation.

## Quantification of astrocyte-specific inflammatory markers in the substantia nigra pars compacta

Formalin-fixed, paraffin-embedded 10 µm brain sections were immunofluorescently-labeled using a Leica Bond RXM automated robotic staining system. Sections were immunohisto-chemically stained on a Leica Bond-III IHC automated stainer. Antigen retrieval was

performed with Bond Epitope Retrieval Solution 2 for 20 minutes. Sections were then incubated with primary antibodies for S100β+ (Abcam; rabbit, 1:500), complement C3 (Abcam; rat monoclonal, 1:100), SerpinA3 (Thermos; mouse monoclonal, 1:250), IP-10/CXCL10 (Santa Cruz; mouse monoclonal, 1:250) and CCL2 (Millipore; mouse monoclonal, 1:500). Secondary antibodies included AlexaFluor anti-rabbit IgG AlexaFluor 488, anti-rat IgG AlexaFluor 555 and anti-mouse AlexaFluor 647. Whole-brain immunofluorescence montage images of labeled tissue sections were imaged using an automated Olympus BX51 fluorescence microscope equipped with a Hamamatsu ORCA-flash 4.0 LT CCD camera and collected using Olympus Cellsens software (v 1.15). Quantitative analysis was performed on dual- or triple-labeled fluorescent images generated by montage imaging of an entire coronal mouse brain section compiled from individual images acquired using an Olympus Plan Apochromat 20X air objective (0.40 N.A.). All slides were scanned under the same conditions for acquisition time, magnification, exposure time, lamp intensity and camera gain. The substantia nigra was delineated by neuroanatomical landmarks and referenced to the Allen brain atlas, following application of an adaptive threshold with shape factor and area ($\mu m^2$) object filters for automatic S100β+ astrocyte cell detection. The number of cells was divided over the area ($\mu m^2$) of the region. To measure expression of inflammatory proteins within S100β+ astrocytes, mean intensities of complement C3, serpinA3, IP-10/CXCL10 and CCL2 were measured by generating automated individual ROIs around all S100β+ astrocytes within the SNpc. We assessed the presence of the inflammatory molecules on two coronal sections per animal that were 10 μm in thickness, spaced at 200 μm intervals within the SNpc, with an $N = 6$–8 mice for each treatment group. The investigator was blinded from all experimental groups during imaging and cell quantitation.

## Quantification of histone acetylation in dopaminergic neurons, microglia and astrocytes in the substantia nigra par compacta

Formalin-fixed, paraffin-embedded 10 μm brain sections were immunofluorescently-labeled as described above using a Leica Bond RXM automated robotic staining system. Sections were then incubated with primary antibodies for tyrosine hydroxylase (WAKO; goat, 1:500), S100β (Abcam; rabbit, 1:500), IBA1 (Abcam; goat, 1:100) and lycine-acetylated histones (Cell Signaling; rabbit, 1:500). Secondary antibodies included anti-goat IgG AlexaFluor 647, anti-mouse IgG AlexaFluor 488, and anti-rabbit IgG AlexaFluor 555. To measure total lysine acetylation in dopaminergic neurons, astrocytes and microglia in the SNpc, labeled tissue sections were imaged using a Olympus Plan Apochromat 20X air objective (0.40 N.A.) and an automated Olympus BX51 fluorescence microscope equipped with a Hamamatsu ORCA-flash 4.0 LT CCD camera and collected using Olympus Cellsens software (v 1.15). Quantitative analysis was performed as described above based on triple-labeled fluorescent images montage images of an entire coronal mouse brain section. All slides were scanned under the same conditions for acquisition time, magnification, exposure time, lamp intensity and camera gain. The substantia nigra was delineated by neuroanatomical landmarks and referenced to the Allen brain atlas, following application of an adaptive threshold with shape factor and area ($\mu m^2$) object filters for automatic TH+ dopaminergic neurons, IBA1+ microglia, and S100β+ astrocyte cell detection. Mean intensities of acetylated lysine residues were measured within TH+ dopaminergic neurons, IBA1+ microglia, and S100β+ astrocytes in the SNpc by generating automated ROIs around each cell type. We assessed total lysine histone acetylation in dopaminergic neurons, astrocytes and microglia in two coronal sections per animal that were 10 μm in thickness, spaced at 200 μm intervals within the SNpc, with an $N = 6$–8 mice for each treatment group. The investigator was blinded from all experimental groups during imaging and cell quantitation.

## RNA sequencing of the substantia nigra

This method was adapted from a previous protocol [32]. A systems-based transcriptional analysis of substantia nigra brain tissues from each treatment group was performed, and the treatment conditions included control, H1N1, and Mn+H1N1. Tissue was anatomically dissected from the basal midbrain and included the substantia nigra but not the ventral tegmental area. Isolated tissue was immediately flash frozen in liquid nitrogen and later homogenized in Trizol reagent (Thermo Fisher) for purification. Samples were then treated with DNAse (Fermentas, Burlington, Ontario) for 30 minutes and purified by phenol/chloroform/isoamyl alcohol (25:24:1) (Fisher Scientific, Pitts- burgh, PA) extraction and ammonium acetate precipitation. Quality and integrity of total RNA was assessed using the 4200 Agilent Tapestation, and samples were confirmed to have RIN scores of >7. RNA [transcripts & non-coding RNA] was isolated from total host RNA followed by library construction and template preparation with the Ion Total RNA-Seq kit and Ion Chef system kit. Sample libraries were prepared using the Ion Total RNA-Seq kit v2 (Life Technologies) and multiplexed on a P1 chip using Ionxpress RNA-Seq 1–16 kit (Life Technologies). Whole mouse transcriptome sequencing was then performed using the Ion Proton Next Generation Sequencer (Life Technologies) through the core facility at the Infectious Disease Research Complex (IDRC) at Colorado State University. Following Next Generation RNA-seq, we used read count coverage (RPKM or FPKM values) to compare the differential gene expression between groups. Advanced RNA-seq analysis was done using the Tuxedo package in Linux command line, including alignments using Bowtie2 and differential gene expression analysis using Cufflinks. Local realignment and base quality score recalibration (BQSR) methods were used as needed to reduce false-positive base calls and improve alignments.

## Analysis of next-generation sequencing data

Data analysis was performed as previously published [32]. In brief, FASTQ files were analyzed using Galaxy for quality trimming, with minimum PHRED quality threshold set at 20 and all read length greater than 20bp. Trimmed reads were then aligned to Mus musculus mm9 using Bowtie2 and gene expression determined using Cufflinks. Expression output was normalized in FPKM format (fragments per kilobase of exon per million reads). Replicate mean values were calculated, and the data was further reduced to FPKM values greater than two. Venn diagrams and Pie chats were generated by comparing the reduced FPKM transcript totals for each treatment group. PANTHER (Protein Analysis Through Evolutionary Relations) Classification System (http://pantherdb.org/publications.jsp#HowToCitePANTHER) was used to ascertain functional pathways driving differences in gene expression by analyzing the complete gene lists within each treatment group for their corresponding annotations, accessed from the Gene Ontology (GO) Consortium. Secondarily, the PANTHER statistical overrepresentation test enabled a comparison of those GO annotations across each treatment group to identify functionally related genes relevant to neurodegeneration. The PANTHER gene list analysis was used to perform a functional classification of all the GO-annotated transcripts within each treatment group. This process employed the GO Term of Molecular Function and GO-slim annotation data sets to analyze the expression profiles of each gene list. We charted any differences between treatment groups as the number of gene hits (the % of gene hits for a GO term /total number of annotated genes in that category). The only GO Term categories shown are those with demonstrable differences in the number of gene hits between the treatment groups. The exact genes returned within each GO classification were analyzed in Excel using MATCH function and dual comparisons to find the specific transcripts unique to each treatment category. To further classify RNAseq transcripts according to function, gene lists were analyzed

using the PANTHER overrepresentation test. This yielded statistically over or under-represented annotations among gene lists relative to the GO Biological Process and Molecular Function *Mus musculus* datasets. Complete gene lists for each treatment group were imported into the analysis tool. Fisher's Exact test for significance was used with FDR multiple test correction. Resulting GO Terms with an enrichment score >1.5 and a FDR<0.05 were considered significant. Genes relevant to neurodegeneration were selected from significant GO Term gene lists. The Search Tool for the Retrieval of Interacting Genes database (STRING) was used to create a network diagram of functional associations between protein products of unique genes within the H1N1+Mn treatment group. Each protein-protein association is weighted according to evidence channels delineated by differing colors. The confidence cutoff was set to 0.4.

## Statistical analysis

All data was presented as mean +/- SEM, unless otherwise noted. Experimental values from each mean were analyzed with a Grubb's ($\alpha = 0.05$) test for exclusion of significant outliers. Differences between each experimental group were analyzed by a one-way ANOVA following a Tukey *post hoc* multiple comparisons test. Significance was identified as [++]$P < 0.01$, [*]$P < 0.05$, [**]$P < 0.01$, [***]$P < 0.001$, [****]$P < 0.0001$. All statistical analysis was conducted using Prism (version 6.0; Graph Pad Software, San Diego, CA).

## Results

### Juvenile manganese exposure increases the number of activated microglial following encephalitic infection with H1N1 in the SNpc

To assess if Mn exposure during juvenile development would enhance the neurological effects of infection with H1N1 in the substania nigra, three week old C57BL/6 mice were administered $MnCl_2$ (50mg/kg/day) or normal drinking water for a total of 30 days and then intranasally infected with H1N1 or mock-infected with saline at 3 months of age (Fig 1A). Stereological determination of TH+ dopaminergic neurons and morphological analysis of Iba1 + microglia at 21 DPI revealed that pre-treatment with $MnCl_2$ during juvenile development induced persistent morphological changes in microglia consistent with an activated phenotype and increased their reactivity to a subsequent infection with H1N1, characterized by retraction of cytoplasmic processes and adoption of an amoeboid phenotype (Fig 1B–1I). We did not observe significant changes in the number of dopaminergic neurons, α-synuclein protein aggregation, or the total number of resting microglia in the SNpc at 21 DPI (Fig 1F and 1G).

### Juvenile manganese exposure increases neuroinflammatory activation of astrocytes following encephalitic infection with H1N1 in the SNpc

Given the increased number of reactive microglia in the substantia nigra and previous work showing that astrocytes play a significant role in microglial activation through glial-glial communication [7, 22, 23, 33, 34], we examined the extent and severity of astrocyte activation in the basal ganglia following treatment with $MnCl_2$ (50mg/kg/day) and intranasal infection with H1N1 at 21 DPI (Fig 2). To determine if dual treatment with Mn enhanced the inflammatory phenotype of astrocytes and increased the number of A1 neurotoxic astrocytes in the SNpc, we measured the level of astrogliosis (Fig 2A–2E), as well as the expression of A1-specific astrocyte inflammatory markers (C3, SerpinA3, IP10, CCL2) in S100β+ astrocytes with immunofluorescence co-localization (Fig 2F–2Y). At 21 DPI, there was a significant increase in S100β+ astrocytes in the SNpc with dual treatment compared to control and Mn treatment alone (Fig 2E).

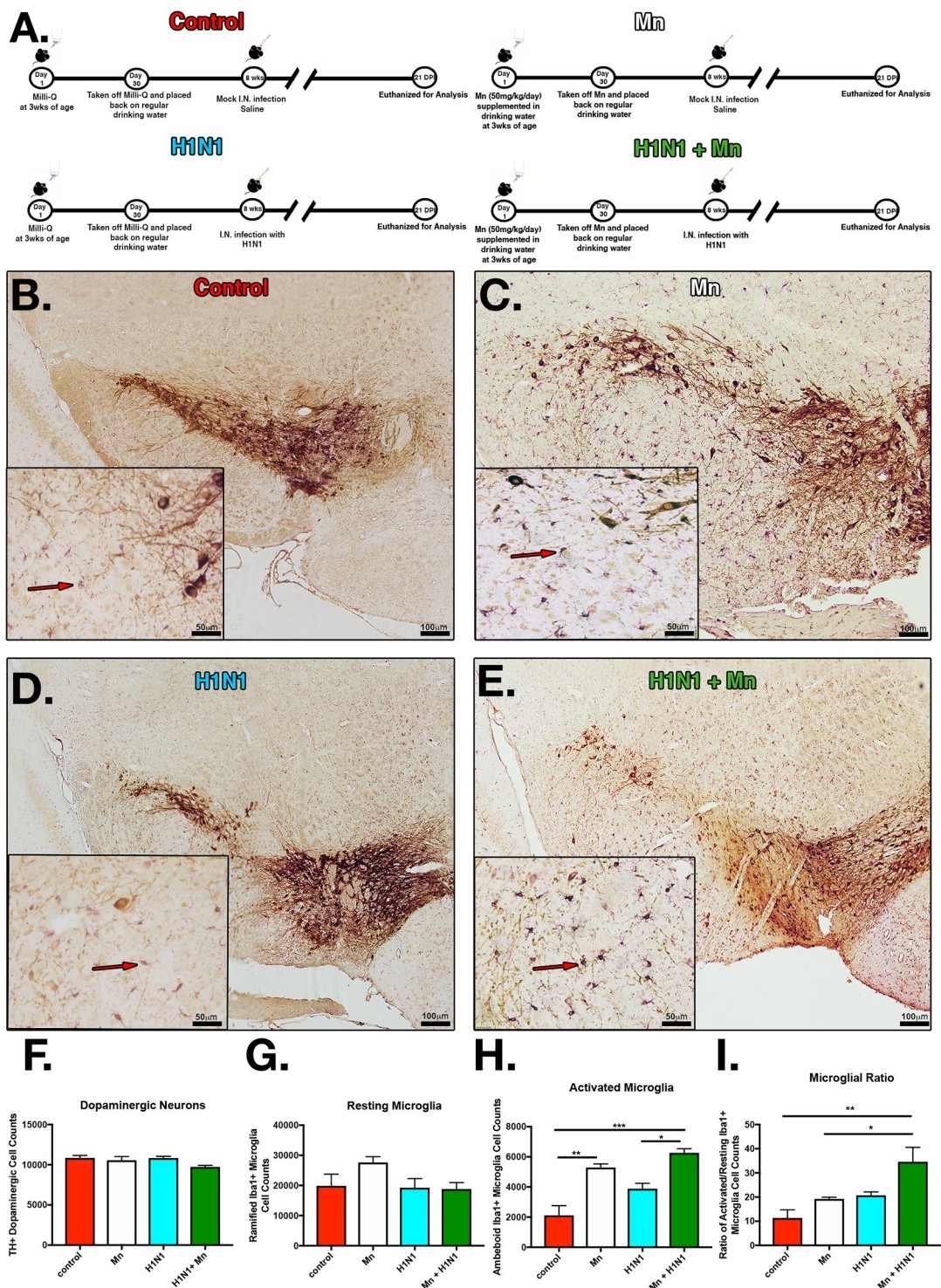

**Fig 1. Pre-treatment with manganese increases microglial activation in the substantia nigra following infection with H1N1.** (A) C57BL/6 mice were divided into groups with and without Mn in drinking water (50 mg/Kg) from day 21–51 PN and then exposed to H1N1 influenza virus three weeks later. Groups: Control, Mn, H1N1, H1N1+MnCl$_2$. (B-E) IHC labeling of dopamine neurons (TH) and microglia (IBA1). (F) Stereological determination of the number of TH+ neurons in the substantia nigra pars compacta (SNpc). (G) Resting microglia in the SNpc. (H) Activated microglia in the SNpc. (I) Ration of activated/resting microglia in the SNpc. $^*P<0.05$ $^{**}P<0.01$ $^{***}P<0.001$$^{****}P<0.0001$. $n$ = 6 mice/group.

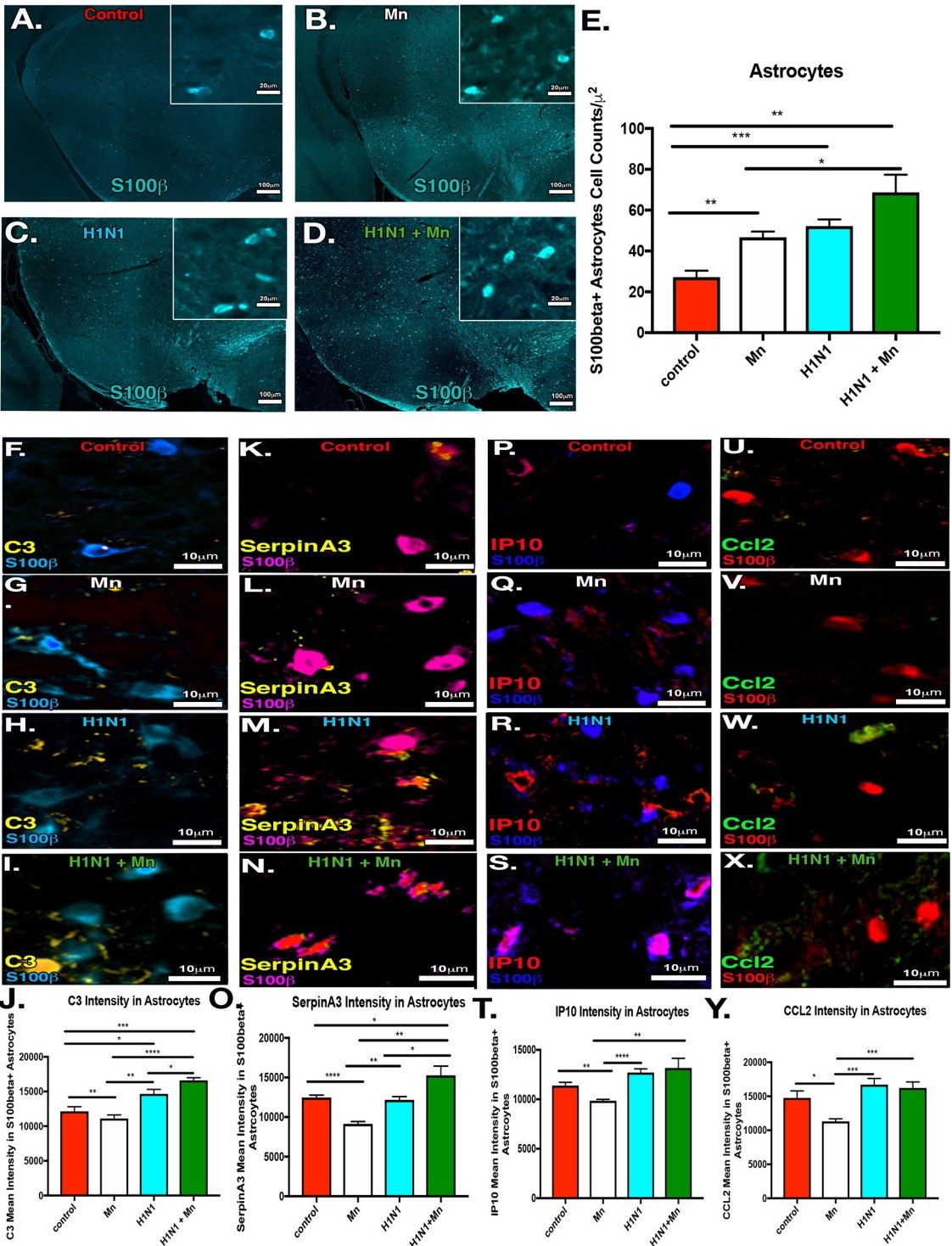

**Fig 2. Pre-treatment with manganese induces proliferation and inflammatory activation in astrocytes in the substantia nigra following infection with H1N1.** (A-D) IF labeling of astrocytes (s100β). Groups: Control, Mn, H1N1, H1N1+ MnCl$_2$. (E) Cell counts of the s100β+ astrocytes in the substantia nigra pars compacta (SNpc). (F-I) IF colocalization of astrocytes (s100β) with complement C3 (C3). (J) C3 mean intensity measurements in s100β+ astrocytes in the SNpc. (K-N) IF colocalization of astrocytes (s100β) with SerpinA3. (O) SerpinA3 mean intensity measurements in S100β+ astrocytes in the SNpc. (P-S) IF colocalization of astrocytes (s100β) with complement IP-10/Cxcl10 (IP-10). (T) IP-10 mean intensity measurements in s100β+ astrocytes in the SNpc. (U-X) IF colocalization of astrocytes (s100β) with complement Ccl2. (J) Ccl2 mean intensity measurements in s100β+ astrocytes in the SNpc.
*$P<0.05$ **$P<0.01$ ***$P<0.001$****$P<0.0001$. $n$ = 6 mice/group.

Additionally, complement C3 and SerpinA3 were significantly increased following pre-treatment with Mn and H1N1 infection compared to control, Mn and H1N1 treatment alone (Fig 2J–2Y). Dual treatment did not induce any differences in expression of Interferon gamma-induced protein 10 (IP-10) or monocyte chemotactic protein (CCL2) (Fig 2P–2Y).

## Dual treatment with Mn and H1N1 alters of histone acetylation in the substantia nigra

To examine the basis for the heightened innate immune inflammatory response observed in glial cells in the basal ganglia following the two-hit exposure model with Mn and H1N1, we assessed histone acetylation in glia and neurons in the substantia nigra. To quantitate the level of cell type-specific histone acetylation in the basal midbrain, brain sections were stained for total acetylated histone lysine residues in TH+ dopaminergic neurons (Fig 3A–3E), S100β⁺ astrocytes (Fig 3F–3J), and IBA1+ microglia in the SNpc (Fig 3K–3O). Infection with H1N1 alone or following Mn pre-treatment significantly decreased histone lysine acetylation in dopaminergic neurons in the SNpc (Fig 3A–3E). Minimal differences were noted in histone acetylation in astrocytes (Fig 3F–3J) In contrast, mice infected with H1N1 alone or following pre-treatment with Mn had significantly increased histone acetylation in IBA1+ microglia in the SNpc (Fig 3F–3O), consistent with previous findings in human PD brains [35].

## Dual treatment with Mn and H1N1 induces a unique transcriptional signature in the substantia nigra consistent with a neurodegenerative phenotype

Given the differences in patterns of histone acetylation in the SNpc following treatment with Mn and H1N1, we performed Next Generation RNA sequencing (RNAseq) of brain tissue from the SN to assess global transcriptional patterns in control mice and those infected with H1N1 as adults with and without Mn pre-treatment during juvenile development (Fig 3). Given our previous findings showing minimal neuropathology in adult mice following treatment with Mn during juvenile development [18, 24, 25], we directly compared H1N1 infected mice to those infected with H1N1 + Mn in the RNA-seq analysis, with both groups referenced to untreated control mice receiving only mock viral infection. Thus, the strategy was to determine how pretreatment with Mn would modulate the severity of infection with H1N1 with respect to changes in gene expression in the SN. RNA was isolated from the SN at 21 DPI for sequencing. The resulting FASTQ files were analyzed using Galaxy for quality trimming, with minimum PHRED quality threshold set at 20 and all read length greater than 20 bp. Trimmed reads were then aligned to the mouse genome using Bowtie2 and gene expression determined using Cufflinks. Local realignment and base quality score recalibration (BQSR) methods were used as needed to reduce false-positive base calls and improve alignments. The expression output was normalized in FPKM format. The replicate mean values were calculated, and the data was further reduced to FPKM values greater than two.

This unbiased global analysis of the transcriptional profile between Control, H1N1 and H1N1+Mn treatment groups revealed that of the 13,168 transcripts annotated from the SN, 351 where unique to control, 290 were unique to H1N1 and 951 transcripts were unique to the dual treatment group (Fig 4A). The complete list of annotated transcripts is provided in S1 Table. To assess whether the increased inflammatory phenotype of microglia and astrocytes in the dual treatment group was a gain of function or loss of function, we next determined the major biological pathways that were altered in each treatment group by using the Gene Ontology (GO) Consortium and PANTHER Classification System pathway and overrepresentation analyses. Resulting transcript annotations were quantitatively different between treatment

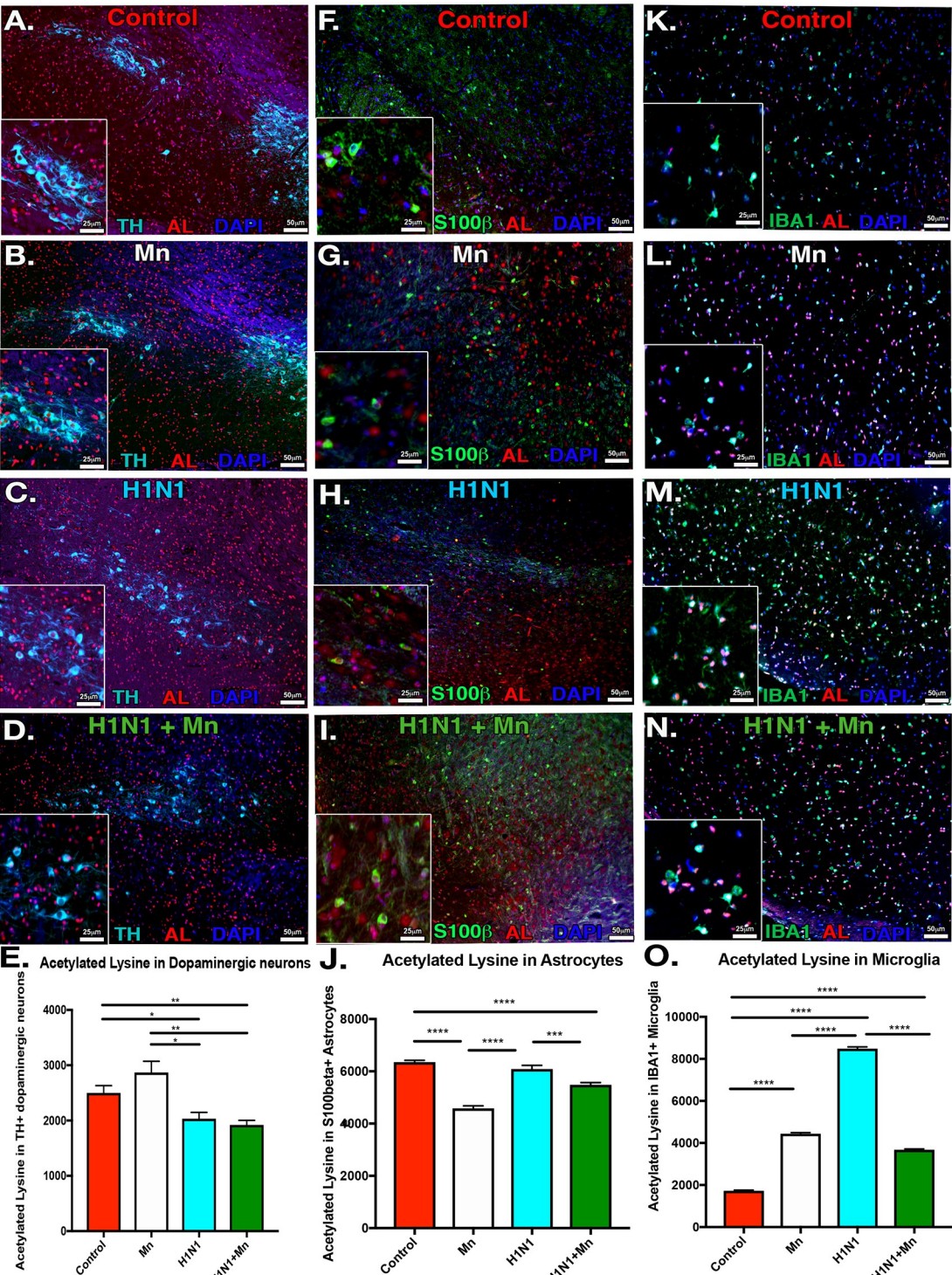

**Fig 3. Dual treatment with MnCl₂ and H1N1 alters histone acetylation in dopaminergic neurons, microglia, and astrocytes in the SNpc.** (A-D) Immunofluorescence co-localization of dopaminergic neurons (TH) and total acetylated lysine residues (AL). Groups: Control, Mn, H1N1, H1N1+ MnCl₂. (E) AL mean intensity measurements in TH+ dopaminergic neurons in the substantia nigra pars compacta (SNpc). (F-G) IF colocalization of astrocytes (s100β) with AL. (J) AL mean intensity measurements in s100β+ astrocytes in SNpc. (K-N) IF colocalization of microglia (Iba1) with AL. (O) AL mean intensity measurements in Iba1+ microglia in the SNpc. *P<0.05 **P<0.01 ***P<0.001****P<0.0001. n = 6 mice/group.

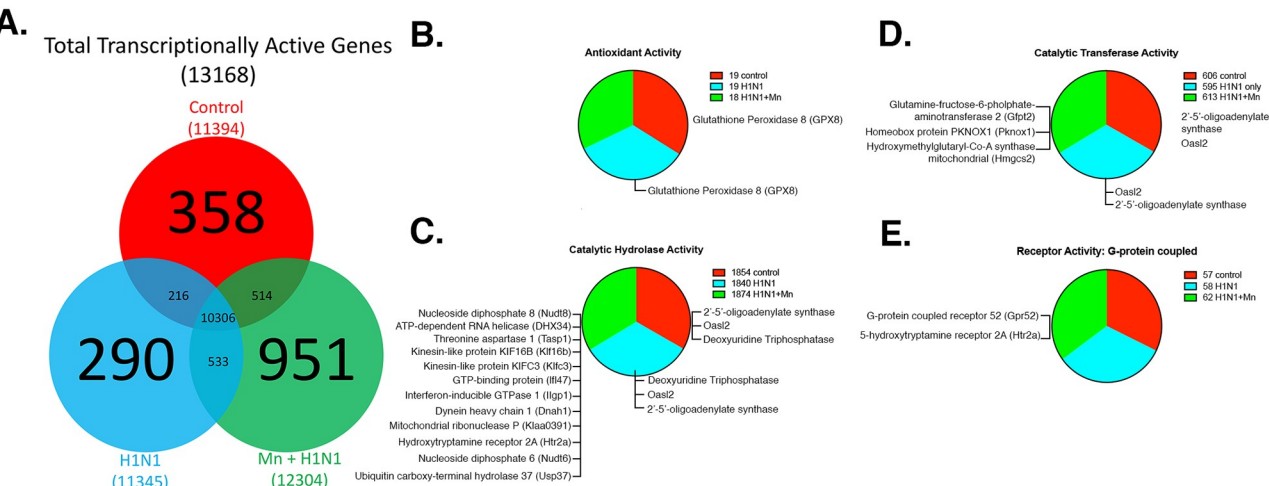

**Fig 4. RNA sequencing of the H1N1+MnCl₂ treatment group highlights uniquely transcribed genes annotated to functionally relevant biological processes.** (A) Venn diagram depicting unique and overlapping transcripts. (B-E) GO Consortium & PANTHER Classification System pathway analysis: Pie charts describe the relative proportions of RNAseq transcripts annotated to each GO Term within the *molecular function* Gene Ontology. The exact gene lists unique or uniquely absent from the H1N1+ Mn treatment group are listed for each category. Genes were classified according to molecular function and biological process with Panther overrepresentation and Functional Enrichment analyses. Groups: Control, H1N1, H1N1+ Mn.

groups in the following four Molecular Function GO Term categories: antioxidant activity (GO:0016209) (Fig 4A and 4B), catalytic hydrolase activity (GO:0016788,GO:0016462, GO:0003924) (Fig 4C), catalytic transferase activity (GO:0008168,GO:0016757,GO:0016301) (Fig 4D), and G-protein coupled receptor activity (GO:0004930) (Fig 4E). The exact genes returned within each GO Term classification were further analyzed to find the specific transcripts unique to each treatment type. Unique transcripts are listed in Fig 4B–4E. Of note, the dual-treatment group active transcript list includes Interferon-inducible GTPase1, Dynein heavy chain, homeobox protein, mitochondrial hydrodoxymethylglutaryl-CoA synthase (Fig 4B–4E), as well as DJ-1/Park 7, other interferon regulatory proteins, autophagy related proteins (Atg), amyloid-beta precursor bindings proteins, NFκB related inflammatory proteins and histone acetyltransferases (S2 Table). Additionally, control and H1N1 treated mice uniquely expressed glutathione peroxidase 8, Oasl3, glutathione-s-transferase, thioredoxin and colony-stimulating factor receptor 1 (CSF1R) (Fig 4B–4E), suggesting that specific stress response pathways are activated and protective mechanisms are lost following dual treatment with Mn and H1N1 when compared to Control and H1N1 treatment alone.

*GO PANTHER analyses were used to further classify and compare transcript lists* according to functional pathways. To understand system-level changes in cellular processes induced by pre-exposure to Mn, we interrogated the protein–protein interaction networks in the dual treatment group using the Search Tool for the Retrieval of Interacting Genes database (STRING) (Fig 5A) [36, 37], wherein network nodes represent proteins and lines represent functional associations between those proteins. The color of each line represents the origin and/or type of evidence supporting that protein-protein interaction and the weights of each line correspond to a confidence score for that evidence type (Fig 5A). From a functional perspective, an association can mean direct physical binding, but STRING evaluates each protein-protein interaction according to evidence from seven different classification channels: neighborhood, co-occurrence, co-expression, experiments, textmining, database and fusion. The *neighborhood* category is related by conserved and co-transcribed operons; the *co-occurrence* category is related by phylogenetic distribution of orthologs of all proteins in a given

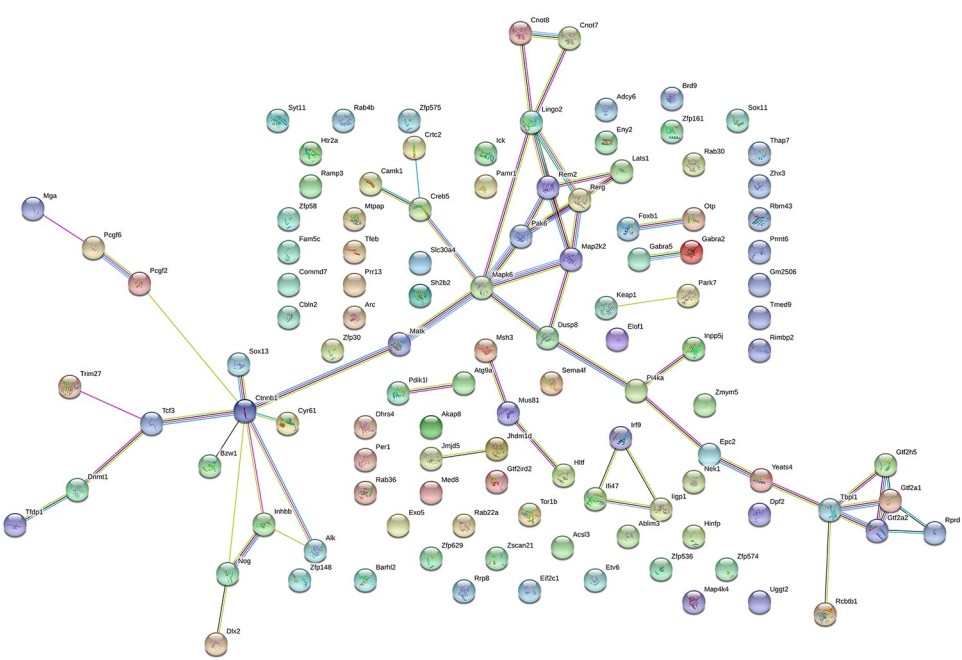

—Neighborhood —Co-occurrence —Co-expression —Experiments —Textmining —Databases —Fusion

**Fig 5. Transcriptional connectome of dual treatment group with associated gene products.** (A) Search Tool for the Retrieval of Interacting Genes database (STRING) diagram of functional associations between protein products of unique genes within the H1N1+Mn treatment group. Nodes represent proteins and edges correspond to functional interactions. Edge colors differ according to a scored confidence scale based on the extent and type of evidence supporting that particular association. Evidence channels: Green Line: Neighborhood, Navy Line: Co-occurrence, Black line: Co-expression, Pink line: Experiments, Yellow line: Textmining, Light blue line: Databases, Red line: Fusion.

organism; the *co-expression* category is related by predicted association between genes based on observed patterns of simultaneous expression of genes; the *experiments* category is related by known experimental interactions; the *textmining* category conducts statistical co-citation analysis across a large number of scientific texts, including all PubMed abstracts and OMIM; the database category is expertly curated and imported from pathway databases; and the *fusion* category is related by proteins that are fused in some genomes and are most likely to be functionally linked [2, 36, 37]. The more lines shown between each protein-protein interaction represents a more likely biological interaction. The most highly represented interactions in the current study were centralized around Lingo 2, Pak6, Tbpl1 and Ctnnb1 (Fig 5A). Analyzing the RNAseq data using STRING methods, we identified a list of genes unique to the H1N1+Mn treatment group that are relevant to neurodegeneration (Table 1). Notable pathways include those contributing to multiple stress responses such as neuroinflammation, oxidative stress, protein misfolding and neurodegeneration. These included anti-oxidant genes such as Kelch-like ECH-associated protein 1 (Keap1), the familial PD gene, Park7, and mitogen activated protein kinase 4k4 (Map4k4), which is involved in the response to environmental stressors and inflammatory cytokines. Other genes at highly interactive nodes unique to the H1N1 + Mn group included Sox11 and 13, which are involved in neurogenesis, and the eukaryotic translation initiation factor 2C1 (Eif2c1), which is involved in protein synthesis and stress responses through post-trasncriptional silencing and damage repair pathways.

**Table 1. H1N1+Mn unique genes relevant to neurodegeneration.**

| Genes | Protein Product description | Pathway of Interest | References |
|---|---|---|---|
| Park7 | Parkinson disease (autosomal recessive, early onset) 7; Protein deglycase that repairs methylglyoxal- and glyoxal-glycated amino acids and proteins and releases repaired proteins and lactate or glycolate, respectively. | SUMOylation of transcription cofactors | [59–62] |
| Lingo2 | Leucine rich repeat and Ig domain containing 2. Genetic polymorphisms in LINGO1 and LINGO2 associated with increased risk of developing essential tremor and Parkinson Disease (PD). Lingo (1,2) is an axonal inhibitor. | Axonal growth inhibition (RHOA activation) | [56, 57, 63] |
| Pak6 | Serine/threonine protein kinase that plays a role in the regulation of gene transcription. The kinase activity is induced by various effectors including AR or MAP2K6/MAPKK6. May protect cells from apoptosis through phosphorylation of BAD. | Activation of RAC1 | [64–66] |
| Eif2c1 | Eukaryotic translation initiation factor 2C1; Required for RNA-mediated gene silencing (RNAi). Binds to short RNAs such as microRNAs (miRNAs) or short interfering RNAs (siRNAs), and represses the translation of mRNAs which are complementary to them. Recent evidence indicates that small RNAs participate in transcriptional regulation in addition to post-transcriptional silencing and damage repair. | Regulation of pTEN mRNA translation | [67–76] |
| Keap1 | Kelch-like ECH-associated protein 1; Acts as a substrate adapter protein for the E3 ubiquitin ligase complex formed by CUL3 and RBX1 and targets NFE2L2/NRF2 for ubiquitination and degradation by the proteasome, resulting in the suppression of its transcriptional activity and the repression of antioxidant response element-mediated detoxifying enzyme gene expression. Retains NFE2L2/NRF2 and may also retain BPTF in the cytosol. Targets PGAM5 for ubiquitination and degradation by the proteasome. | Ub-specific processing proteases | [77–80] |
| Sox11 | Transcriptional factor involved in the embryonic neurogenesis. May also have a role in tissue modeling during development. | Binding of chemokine receptors | [81–84] |
| Sox13 | Member of SOX family of transcription factors. | Binding of chemokine receptors | [81–84] |
| Arc | Activity regulated cytoskeletal-associated protein; Plays a role in the regulation of cell morphology and cytoskeletal organization. Required in the stress fiber dynamics, cell migration, consolidation of synaptic plasticity and formation of long-term memory. | Trafficking of AMPA receptors | [85–87] |
| Iigp1 | GTPase with low activity. Has higher affinity for GDP than for GTP. Plays a role in resistance to intracellular pathogens. Mediates resistance to infection by targeting bacterial inclusions to autophagosomes for subsequent lysosomal destruction. | Resistance to infection. | [88] |
| Camk1 | Calcium/calmodulin-dependent protein kinase that operates in the calcium-triggered CaMKK-CaMK1 signaling cascade and, upon calcium influx, regulates transcription activators activity, cell cycle, hormone production, cell differentiation, actin filament organization and neurite outgrowth. | Transcriptional activation of mitochondrial biogenesis. | [89–91] |
| Slc30a4 | Likely involved in zinc transport out of the cytoplasm, perhaps be by sequestration into an intracellular compartment. | Zinc efflux | [92–94] |
| Atg9a | Involved in autophagy and cytoplasm to vacuole transport (Cvt) vesicle formation. Plays a key role in the organization of the preautophagosomal structure/phagophore assembly site (PAS), the nucleating site for formation of the sequestering vesicle. | Macroautophagy | [95–98] |
| Per1 | Transcriptional repressor that forms a core component of the circadian clock. | Circadian clock | [99–103] |
| Gabra2 | Gamma-aminobutyric acid (GABA) A receptor, subunit alpha 2; GABA, the major inhibitory neurotransmitter in the vertebrate brain, mediates neuronal inhibition by binding to the GABA/benzodiazepine receptor and opening an integral chloride channel. | Neurotransmitter receptors and postsynaptic signal transmission | [99–103] |
| Gabra5 | Gamma-aminobutyric acid (GABA) A receptor, subunit alpha 5; GABA, the major inhibitory neurotransmitter in the vertebrate brain, mediates neuronal inhibition by binding to the GABA/benzodiazepine receptor and opening an integral chloride channel. | Neurotransmitter receptors and postsynaptic signal transmission | [104–106] |
| Ildr2 | Immunoglobulin-like domain containing receptor 2; May be involved in ER stress and lipid homeostasis. | ER stress pathway | [107–110] |
| Tcf3 | Transcription factor 3; Transcriptional regulator. Involved in the initiation of neuronal differentiation. Heterodimers between TCF3 and tissue-specific basic helix-loop-helix (bHLH) proteins play major roles in determining tissue-specific cell fate during embryogenesis, like muscle or early B-cell differentiation. | CDO (cell-adhesion-molecule/downregulated by oncogenes) in myogenesis | [111] |
| Map4k4 | Serine/threonine kinase that may play a role in the response to environmental stress and cytokines such as TNF-alpha. Appears to act upstream of the JUN N-terminal pathway. Phosphorylates SMAD1 on Thr-322. | Oxidative stress induced senescence | [112–121] |

PANTHER overrepresentation analysis was used to more narrowly identify unique genes relevant to neurodegeneration. Results with FDR<0.05 and an enrichment score >1.5 were considered significant. From among statistically significant GO Terms, genes annotated to those categories were selected according to relevance to neurodegeneration.

## Discussion

Although the majority of PD cases are sporadic and of unknown etiology, there is increasing evidence that environmental stressors such as neurotoxic chemicals and encephalitic infections can increase susceptibility to this and other neurological diseases [4, 38, 39]. Chronic inflammatory activation of glia in the nigro-striatal pathway is a well-established feature of both sporadic PD and post-encephalitic parkinsonism and could represent a cellular mechanism linking environmental exposures early to potentiation of neurodegeneration following subsequent insults. Using a two-hit model with H1N1 and MPTP, it was recently shown that influenza infections can enhance innate immune responses of microglia in the SNpc following exposure to the parkinsonian agent, MPTP [16]. It is not known whether chronic exposure to environmental metals such as Mn have a similar capacity to enhance neurological injury from viral infection. However, earlier studies in mice reported that a single dose of Mn or Cd enhanced the neurovirulence of infection with several alphaviruses and increased the severity of symptoms, neuroinflammation and mortality [40]. To test if exposure to Mn during juvenile development could similarly prime glial cells in the SNpc for a more severe neuroinflammatory response following encephalitic infection with H1N1, we utilized a two-hit exposure model with Mn and H1N1 and examined whether there were epigenetic and transcriptomic changes in the SNpc that could explain the heightened innate immune response to viral infection in microglia and astrocytes.

Our results suggest that Mn exposure during juvenile development induces lasting neuroinflammatory and epigenetic alterations in glia that exacerbate the neuroinflammatory response to H1N1 compared to mice that did not receive excess Mn as juveniles. Although we did not observe overt loss of dopaminergic neurons or protein aggregation of α-synuclein at 21 DPI in any treatment group (Fig 1), there was a marked increase in activation of microglia that correlated with a downward trend in the number of dopaminergic neurons that would likely accelerate with aging, thus predisposing to neurological dysfunction. Therefore, these pathological findings may reflect early neuropathological changes corresponding the prodromal stage of PD or viral parkinsonism. This highlights the importance of neurotoxic environmental exposures that promote a reactive inflammatory phenotype in microglia that could predispose to injury within the nigro-striatal dopamine system. Recent studies support a critical role for Mn in modulating innate immunity in response to viral infection through enhancing the sensitivity of pattern recognition receptors that then stimulate anti-viral gene expression [41]. Release of Mn into the cytosol during infection with dsDNA viruses increases activation of the cGAS-STING pathway by elevating production of cGAMP that stimulates NFκB-dependent expression of the anti-viral interferon response. This is consistent with previous data from our laboratory demonstrating that Mn directly stimulates soluble guanylate cyclase and thereby enhances NFκB-induced expression of inflammatory genes through increases in cGMP [25].

Chronic neuroinflammatory activation of microglia and astrocytes is a central feature of aging, viral encephalitis and neurodegenerative disease. Increasing evidence implicates microglia-astrocyte signaling in neuroinflammatory responses that enhance neurodegeneration [42–45]. Our lab and others have demonstrated that microgliosis precedes astrogliosis and neuronal loss in models of PD [22, 46, 47], indicating that cytokine and chemokine signaling from neuroinflammatory activation of microglia likely plays a critical role in inflammatory activation of astrocytes in the SNpc. Given that dual treatment with Mn and H1N1 potentiated inflammatory activation of the A1 astrocyte markers, SerpinA3 and C3 (Fig 2), and also increased the number of reactive microglia (Fig 1), these results provide additional evidence that innate immune signaling in activated microglia plays a critical role in inflammatory activation of astrocytes. Mn directly stimulates expression of TNF in microglia that can enhance

the production of inflammatory cytokines and chemokines in astrocytes [22], including CCL2, that promote neuronal injury [33]. Thus, inflammatory 'priming' of microglia during juvenile exposure to Mn could enhance activation of astrocytes into a neurotoxic A1 phenotype during subsequent exposure to H1N1. The data reported in the current study supports this model, with marked increases in expression of the A1 proteins SerpinA3 and C3 in reactive astrocytes in dual-treated mice (Fig 3). However, Mn pre-exposure did not increase levels of IP10 and CCL2 in astrocytes following H1N1 infection, suggesting that individual neurotoxic exposures or pathogens likely have a unique molecular signature associated with distinct inflammatory phenotypes in astrocytes.

The neurotoxic effects of Mn are mediated both through direct toxic effects in neurons, as well as through activation of inflammatory signaling pathways which further injure neurons through overproduction of reactive oxygen and nitrogen species and inflammatory cytokines [19]. Thus, the capacity of environmental exposure to sensitize neural tissue to additional damage from subsequent H1N1 encephalitic infection may involve persistent inflammatory changes in glial cells [14]. It has previously been shown that juvenile exposure to Mn can induce lasting effects on the neuroinflammatory status of astrocytes and microglia which continues into adulthood and this likely contributes increased susceptibility to secondary environmental insults and infections [18, 24, 25, 48, 49]. However, the mechanisms regulating this sustained inflammatory state after environmental insults in microglia and astrocytes are unclear. Previous studies have reported that manganese can alter histone acetylation and gene expression, chromatin remodeling, cell cycle progression, DNA repair and apoptosis in neurons and glia [12, 50]. Glial cells possess innate immune memory to environmental stimuli through histone acetylation that alters subsequent inflammatory responses [12]. Histone acetylation is also altered in PD, likely associated with microglial activation in the SNpc that increases dopaminergic neurodegeneration [35, 51, 52]. Additionally, decreased acetylation in TH neurons has been noted in clinical PD cases, and histone acetylation is thought to be modulated in glia following exposure to other environmental insults associated with PD [35, 51–53]. Here we observed that juvenile exposure to Mn followed by adult infection with H1N1 causes changes in histone acetylation in dopaminergic neurons, microglia and astrocytes in the SNpc, relative to infection with H1N1 alone (Fig 3). Also, dual treatment with Mn and H1N1 decreased histone acetylation in dopaminergic neurons, corresponding with data in clinical cases of PD [35, 51–53]. We surmise that these epigenetic changes in glia likely play a role in their increased inflammatory activation to infection with H1N1. However, whether these epigenetic changes persist throughout aging remains to be determined.

It is still unclear whether neuroinflammatory activation of glia increases susceptibility to neurodegeneration primarily through decreased release of neurotrophic factors or from excessive synthesis of neurotoxic inflammatory mediators [47, 54]. Glial activation is regulated through multiple pathways including mitogen-activated protein kinases (MAPKs), activator protein-1 (AP-1), Janus kinase (JAK)/signal transducer and activator of transcription (STAT), interferon regulatory factor families (IFN), as well as through the nuclear factor kappa B (NFκB) pathway. To address this question, we performed Next Generation RNA-sequencing to examine gene networks regulated by Mn that could predispose neurons in the substantia nigra to injury following infection with H1N1. We found that pre-treatment with Mn prior to infection with H1N1 increased the number of unique transcripts and significantly altered the global transcriptional profile in the SNpc compared to H1N1 treatment alone (Fig 4). Amongst unique transcripts in the dual treatment group, STRING analysis revealed an overrepresentation of Interferon-inducible GTPase1, dynein heavy chain, homeobox protein, mitochondrial hydrodoxymethylglutaryl-CoA synthase (Fig 4B–4E), as well as DJ-1/Park 7, other interferon regulatory proteins, autophagy-related proteins (Atg), amyloid-beta precursor binding

proteins, NFκB related inflammatory proteins and histone acetyltransferases (S2 Table). Park7 (DJ1) is linked to the preservation of mitochondrial function in PD, and the unique activation of DJ-1, Atg and interferon regulatory protein in the dual treatment group likely represents the activation of the antioxidant activation and autophagy stress response pathways [55], possibly in reponse to underrepresented transcripts involved in antioxidant activity, such as glutathione peroxidase (Fig 4A). These data suggest that increased inflammatory activation of glia in the SNpc in the Mn + H1N1 group was associated with stress reponses in this brain region consistent with inflammation and innate immune function, as well as oxidative stress and mitophagy. Moreover, given that viral infections can inhibit mitochondrial bioenergetics by depressing cellular ATP content and induce oxidative stress in neurons and glia, these data suggest that juvenile exposure to Mn could reduce glutathione-dependent anti-oxidant protection through a reduction of glutathione peroxidase that exacerbates oxidative stress during subsequent infection with H1N1 [42].

Analysis of RNA sequencing data also identified unique patterns of gene expression within each treatment group. Annotated of genes to specific molecular function and biological processes (Fig 4) enabled us to generate a network summarizing predicted associations between gene products to attain a better systems-level understanding of cellular processes in the dual treatment group (Fig 5). Dj-1/Park7, Lingo2, Keap1, Eif2c1, and Pak6 were all unique genes identified using overrepresentation analysis that are relevant to etiology and/or progression of PD (Table 1). Interestingly, LINGO2 is a member of LRR gene family that, along with LRRK2, has been linked to Essential tremor (ET) and PD and has even become a promising therapeutic target in multiple sclerosis (MS) and PD [56–58]. Taken together, these findings support the involvement of distinct proteins and pathways in the neurologically deleterious effects caused by successive environmental challenge with Mn and H1N1. Although there was not significant loss of dopaminergic neurons at the relatively early timepoint evaluated, the unique protein-protein interactions in dual treated animals could represent an early pre-symptomatic stage of neuronal dysfunction corresponding to prodromal disease.

The present studies used a two-hit model of successive exposure to Mn and H1N1 influenza virus to identify mechanisms by which multiple environmental insults and microbial infection could act in concert to increase susceptibility to PD and related neurodegenerative diseases. Exposure to Mn from PN 21–51 did not result in significant loss of dopaminergic neurons in the SNpc but markedly increased neuroinflammatory activation of microglia and astrocytes that could represent a form of innate immune memory in the brain that predisposes glia cells to a neurotoxic reactive phenotype during aging. This is consistent with the patterns of gene expression noted in RNA-Seq studies that revealed transcriptional signatures consistent with a neurodegenerative phenotype, particularly for genes related to oxidative stress, mitophagy, protein processing, and immune function. The unique patterns of gene expression noted in animals exposed to both Mn and H1N1 suggests that the mechanism by which multiple environmental exposures modulate neurotoxic injury seen in certain cases of sporadic and post-encephalitic PD may involve epigenetic changes that favor expression of genes associated with inflammation and protein misfolding in astrocytes and microglia.

## Supporting information

**S1 Table. Complete list of expressed transcripts in each experimental group.** The complete list of annotated transcripts from unbiased global analysis of the transcriptional profile of Control, H1N1 and H1N1+Mn treatment groups is presented in S1 Table.
(XLSX)

**S2 Table. Uniquely expressed transcripts in each experimental group.** Transcripts uniquely expression in each experimental group are presented in S2 Table. Mapping these transcripts to the major biological pathways that were altered in each treatment group was performed using the Gene Ontology (GO) Consortium and PANTHER Classification System pathway and overrepresentation analyses.
(XLSX)

## Acknowledgments

The authors acknowledge the technical assistance of animal care and use veterinary personnel at Colorado State University and Thomas Jefferson University.

## Author Contributions

**Conceptualization:** Collin M. Bantle, Richard A. Slayden, Richard J. Smeyne, Ronald B. Tjalkens.

**Data curation:** C. Tenley French, Jason E. Cummings, Richard A. Slayden, Richard J. Smeyne, Ronald B. Tjalkens.

**Formal analysis:** Collin M. Bantle, C. Tenley French, Jason E. Cummings, Shankar Sadasivan, Kevin Tran, Richard A. Slayden, Richard J. Smeyne, Ronald B. Tjalkens.

**Funding acquisition:** Ronald B. Tjalkens.

**Investigation:** Collin M. Bantle, Kevin Tran, Richard A. Slayden, Richard J. Smeyne, Ronald B. Tjalkens.

**Methodology:** Shankar Sadasivan, Richard A. Slayden, Richard J. Smeyne, Ronald B. Tjalkens.

**Project administration:** Ronald B. Tjalkens.

**Resources:** Richard J. Smeyne, Ronald B. Tjalkens.

**Supervision:** Ronald B. Tjalkens.

**Validation:** Ronald B. Tjalkens.

**Writing – original draft:** Collin M. Bantle, C. Tenley French, Jason E. Cummings, Richard A. Slayden, Richard J. Smeyne, Ronald B. Tjalkens.

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
