## [Decision Letter · Decision Letter 0]

1 Dec 2020

PONE-D-20-34080

Manganese exposure in juvenile C57BL/6 mice increases glial inflammatory responses in the substantia nigra following infection with H1N1 influenza virus

PLOS ONE

Dear Dr.  Tjalkens,

Thank you for submitting your manuscript to PLOS ONE. After careful consideration, we feel that it has merit but does not fully meet PLOS ONE’s publication criteria as it currently stands. Therefore, we invite you to submit a revised version of the manuscript that addresses the points raised during the review process.

See comments below.

Also, we need clarification if figures 1 and 2 were taking at the same coordinates (X,Y,Z) because the size looks significantly different.

Please submit your revised manuscript in 10-15 days. If you will need more time than this to complete your revisions, please reply to this message or contact the journal office at plosone@plos.org. Please include the following items when submitting your revised manuscript:

We look forward to receiving your revised manuscript.

Kind regards,

Eliseo A Eugenin, Ph.D.

Academic Editor

PLOS ONE

Additional Editor Comments:

Hi Dr. Tjalkens

Thank you for submit your manuscript to PLOSone. Attached are the comments of the reviewers. please add the information requested and submitted back

Best regards

Eliseo

Journal Requirements:

2. In your Methods section, please provide additional details regarding the animals used in your study and ensure you have described the source. For more information regarding PLOS' policy on materials sharing and reporting, see https://journals.plos.org/plosone/s/materials-and-software-sharing#loc-sharing-materials.

3. We note that you are reporting an analysis of a microarray, next-generation sequencing, or deep sequencing data set. PLOS requires that authors comply with field-specific standards for preparation, recording, and deposition of data in repositories appropriate to their field. Please upload these data to a stable, public repository (such as ArrayExpress, Gene Expression Omnibus (GEO), DNA Data Bank of Japan (DDBJ), NCBI GenBank, NCBI Sequence Read Archive, or EMBL Nucleotide Sequence Database (ENA)). In your revised cover letter, please provide the relevant accession numbers that may be used to access these data. For a full list of recommended repositories, see http://journals.plos.org/plosone/s/data-availability#loc-omics or http://journals.plos.org/plosone/s/data-availability#loc-sequencing.

Reviewers' comments:

Reviewer's Responses to Questions

**Comments to the Author**

1. Is the manuscript technically sound, and do the data support the conclusions?

Reviewer #1: Yes

Reviewer #2: Yes

2. Has the statistical analysis been performed appropriately and rigorously? 

Reviewer #1: Yes

Reviewer #2: Yes

3. Have the authors made all data underlying the findings in their manuscript fully available?

Reviewer #1: Yes

Reviewer #2: Yes

4. Is the manuscript presented in an intelligible fashion and written in standard English?

Reviewer #1: Yes

Reviewer #2: Yes

5. Review Comments to the Author

Reviewer #1: Review Comments to the Author

Please use the space provided to explain your answers to the questions above. You may also include additional comments for the author, including concerns about dual publication, research ethics, or publication ethics. (Please upload your review as an attachment if it exceeds 20,000 characters) (Limit 200 to 20000 Characters)

I feel that the authors have written a good manuscript

Reviewer #2: Influenza A viruses affecting birds and mammals, with a high risk of serious illness and death worldwide. While the primary target of influenza viruses in mammals is the lung, neurological complications were also reported, and still is unknown the mechanisms and onsequences of neuroinflammation caused by influenza A viruses are only partly understood. Still unknown how influenza strains are able to enter the central nervous system (through the blood-brain barrier or microvascular endothelial cells, or blood—cerebrospinal fluid barrier).

In the article “Manganese increases the neuroinflammatory effects of H1N1 infection” authors test the hypothesis, that interactions with an environmental neurotoxin could promote more severe neurological damage in surviving patients of H1N1 infections up to the development of Parkinson's. In study C57BL/6 mice were exposed to MnCl 2 in drinking water (50 mg/kg/day) for 30 days from days 21 – 51 postnatal, then infected intranasally with H1N1 three weeks later. Authors showed that dual treatment with Mn and H1N1 potentiated inflammatory activation of the A1 astrocyte markers, SerpinA3 and C3, and also increased the number of reactive microglia, these results provide additional evidence that innate immune signaling in activated microglia plays a critical role in inflammatory activation of astrocytes. Analyses of dopaminergic neurons, microglia and astrocytes in basal ganglia indicated that although there was no significant loss of dopaminergic neurons within the substantia nigra pars compacta, there was more pronounced activation of microglia and astrocytes in animals sequentially exposed to Mn and H1N1. These results add a new information that exposure to elevated levels of Mn during juvenile development could sensitize glial cells to more severe neuro-immune responses to influenza infection later in life through persistent epigenetic changes. In general, the study analyses are performed to a high technical standard and are described in sufficient detail; conclusions are presented in an appropriate fashion and are supported by the data. The article is presented in an intelligible fashion and is written in standard English.

6. PLOS authors have the option to publish the peer review history of their article (what does this mean?). If published, this will include your full peer review and any attached files.

---

## [Author Response · Author response to Decision Letter 0]

21 Dec 2020

PONE-D-20-34080-R1

Response to Reviewer Concerns

1. “Please ensure that your manuscript meets PLOS ONE's style requirements, including those for file naming.”

The manuscript has been carefully revised to insurance compliance with the style requirements for PLoS ONE, including file naming.

2. “In your Methods section, please provide additional details regarding the animals used in your study and ensure you have described the source.”

The Methods have been updated in the revised manuscript (p. 5, paragraph 1) to note the source vendor of the animals to add details on husbandry.

3. “We note that you are reporting an analysis of a microarray, next-generation sequencing, or deep sequencing data set. PLOS requires that authors comply with field-specific standards for preparation, recording, and deposition of data in repositories appropriate to their field.”

Sequence Read Archive (SRA) NCBI accession number added to revised cover letter.

4. “Please include captions for your Supporting Information files at the end of your manuscript, and update any in-text citations to match accordingly."

Captions for supplemental figures are included at the end of manuscript per the Reviewer’s request.

5. "Also, we need clarification if figures 1 and 2 were taking at the same coordinates (X,Y,Z) because the size looks significantly different.”

Yes, Figures 1 and 2 were taken at close proximal anatomical coordinates to obtain analysis of TH+ neurons, microglia and astocytes within the substantia nigra. The immunohistochemical images in Figure 1 provide a slightly different perspective than the immunofluorescence images in Figure 2 but every effort was made to include sections from the same relative anatomical plane through the basal midbrain.

6. Reviewer 2 comment: “These results add a new information that exposure to elevated levels of Mn during juvenile development could sensitize glial cells to more severe neuro-immune responses to influenza infection later in life through persistent epigenetic changes. In general, the study analyses are performed to a high technical standard and are described in sufficient detail; conclusions are presented in an appropriate fashion and are supported by the data.”

We thank the Reviewer for this critique and agree that the finding of neuro-immune priming of microglia by early life exposures to manganese that increases astrocyte reactivity upon challenge with H1N1 is an important finding in the field, especially considering the current global pandemic in which prior exposure to endogenous or exogenous toxins increases virulence. We suggest in the conclusions that this two-hit interaction between environmental toxins and infectious viruses may be important to understanding the etiology of various neurodegenerative diseases.

---

## [Editor Report · Decision Letter 1]

23 Dec 2020

Manganese exposure in juvenile C57BL/6 mice increases glial inflammatory responses in the substantia nigra following infection with H1N1 influenza virus

PONE-D-20-34080R1

Dear Dr.Tjalkens ,

We’re pleased to inform you that your manuscript has been judged scientifically suitable for publication and will be formally accepted for publication once it meets all outstanding technical requirements.

Kind regards,

Eliseo A Eugenin, Ph.D.

Academic Editor

PLOS ONE

Additional Editor Comments (optional):

Dear Dr. Tjalkens

Thank you for answering and clarifying the all the questions. Thank you for submit your manuscript to PLOSone

Best Regards

Eliseo Eugenin
---

## [Editor Report · Acceptance letter]

13 Jan 2021

PONE-D-20-34080R1 

Manganese exposure in juvenile C57BL/6 mice increases glial inflammatory responses in the substantia nigra following infection with H1N1 influenza virus 

Dear Dr. Tjalkens:

I'm pleased to inform you that your manuscript has been deemed suitable for publication in PLOS ONE. Congratulations! Your manuscript is now with our production department. 

Kind regards, 

on behalf of

Dr. Eliseo A Eugenin 

Academic Editor

PLOS ONE